# ITS1 Barcode and Phytochemical Analysis by Gas Chromatography–Mass Spectrometry of *Corynaea crassa* Hook. f (Balanophoraceae) from Ecuador and Peru

**DOI:** 10.3390/genes14010088

**Published:** 2022-12-28

**Authors:** Alexandra López-Barrera, Efrén Santos-Ordóñez, Ricardo Pacheco-Coello, Liliana Villao-Uzho, Migdalia Miranda, Yamilet Gutiérrez, Iván Chóez-Guaranda, Segundo Guillermo Ruiz-Reyes

**Affiliations:** 1Department of Pharmacy, Faculty of Chemical Sciences, Universidad de Guayaquil, Guayaquil 090514, Ecuador; 2Facultad de Ciencias de la Vida, Campus Gustavo Galindo, ESPOL Polytechnic University, Escuela Superior Politécnica del Litoral, ESPOL, Guayaquil 090902, Ecuador; 3Centro de Investigaciones Biotecnológicas del Ecuador, Campus Gustavo Galindo, ESPOL Polytechnic University, Escuela Superior Politécnica del Litoral, ESPOL, Guayaquil 090902, Ecuador; 4Facultad de Ciencias Naturales y Matemáticas, ESPOL Polytechnic University, Escuela Superior Politécnica del Litoral, ESPOL, Guayaquil 090902, Ecuador; 5Department of Pharmacy, Institute of Pharmacy and Food, Universidad de La Habana, La Habana 10400, Cuba; 6Faculty of Pharmacy and Biochemistry, National Trujillo University, Trujillo 13011, Peru

**Keywords:** internal transcribed spacer, medicinal, parasitic plant

## Abstract

The use of medicinal plants is the basis of traditional healthcare. Recently, the use of herbal medicine has been increasing among consumers due to availability, economy, and less side effect. For instance, the hemiparasite plant *Corynaea crassa* has medicinal properties and could be found in some regions of America, from Costa Rica to Bolivia. Phytochemical and genetic characterization of medicinal plants is needed for proper identification of metabolites responsible for medicinal properties and for genotyping, respectively. Moreover, characterization of medicinal plants through the use of DNA barcodes is an important tool for phylogenetic analysis and identification of species; furthermore, complemented with phytochemical analysis, both are useful for identification of plant species and quality control of medicinal products. The objective of this study was to analyze the species of *C. crassa* collected in Ecuador and Peru from the phylogenetic and phytochemical point of view. Polymerase chain reaction (PCR) was performed for amplification of the internal transcribed spacer 1 (ITS1) region after DNA extraction of samples of *C. crassa*. Blast analysis was performed in the GenBank database with the ITS1 sequences obtained from two accessions of *C. crassa* from Ecuador (GenBank accession numbers OM471920 and OM471919 for isolates CIBE-17 and CIBE-18, respectively) and three from Peru (GenBank accession numbers OM471921, OM471922, and OM471923 for isolates CIBE-13, CIBE-14, and CIBE-15, respectively). The accessions available in the GenBank were used for phylogenetic analysis. For the phytochemical analysis, hydroalcoholic extracts were obtained by maceration using 80% ethanol as solvent, followed by a derivatization process and analysis by gas chromatography–mass spectrometry. Based on the phylogenetic analysis of the *C. crassa* samples, the ITS1 sequence could be used to differentiate *C. crassa* of different locations. The samples of *C. crassa* from Ecuador and Peru are more similar between them than with other clades including *Helosis* spp. The phytochemical study revealed differences in the presence and relative abundance of some metabolites; mainly eugenol, 1,4-lactone arabinonic acid, dimethoxyrabelomycin and azelaic acid, which are reported for the first time for the species under study and the genus *Corynaea*. These results are the first findings on the combined analysis using genetic and phytochemical analysis for *C. crassa*, which could be used as a useful tool for quality control of the *C. crassa* species in medicinal products.

## 1. Introduction

The use of medicinal plants is the basis of traditional healthcare. Mainly in developing countries and worldwide, over 80% of the population depends on traditional and herbal medicine [1]. Traditional medicine and modern drug discovery using medicinal plants to treat chronic diseases and maintain world health is well documented [2]. Furthermore, plant-derived products are in at least 25% of drugs in the modern pharmacopeia [1]. Recently, the use of herbal medicine has been increasing among consumers due to availability, economy, and less side effect. At least 30% of commercial drugs used in the pharmaceutical industry derived from natural products [3]. Furthermore, the interest of medicinal plants worldwide is growing.

In Peruvian folk medicine, several medicinal plants are used for aphrodisiacal application. One of the plant ingredients used in these aphrodisiac mixtures is derived from a plant denominated *Chutarpo* or *huanarpo macho*, with scientific name *C. crassa* Hook. f. [4,5]. *C. crassa* is a hemiparasitic plant that belongs to a genus of flowering plants of Balanophoraceae that includes about 45 species. *C. crassa* is distributed in some regions of America, from Costa Rica to Bolivia [6,7]. The morphological studies at the macroscopic and microscopic level of plant drugs are important tools for the identification of similarities and differences between the plants. Differences were observed in the size of the plants of *C. crassa* of Ecuador and Peru, due to different ecological factors that may influence plant development. The macroscopic and microscopic analysis showed no structural changes between the powdered drugs of both *C. crassa* from Ecuador and Peru, except for the contour of epidermal cells that appear more pronounced in the *C. crassa* from Ecuador, suggesting that both are the same species [8].

From the pharmacological point of view, the *C. crassa* hydroalcoholic extracts from Ecuador and Peru demonstrated antioxidant activity, similar or superior to the reference substances vitamin C and Trolox [9]. In addition, the aqueous and hydroalcoholic extracts showed an anti-inflammatory effect in the plantar edema test induced by carrageenan in the rat paw [8]. Furthermore, Bussmann et al. [4] reported that the species had antimicrobial activity. Several chemical compounds have been identified in the species, including terpenes (monoterpenes, triterpenoids), flavonoids (catechin, quercetin glycoside, and a flavanone glycoside), tannins, and fatty acids, among others [5,8,9].

DNA barcoding has been proposed as a complement for taxonomic analysis for species identification and is recommended as an important tool in quality control or adulteration of medicinal plants [10]. DNA barcode is a short DNA sequence which is used for identification of species [10] that could be used for rapid, accurate tool as a complement for taxonomic identification. Several molecular barcodes have been used in medicinal plants, including the chloroplast *rbcL*, *matK* genes, and the nuclear internal transcribed spacers 1 (ITS1) and 2 (ITS2) sequences [11,12]. Therefore, as a complement for species identification, the DNA barcode analysis and bioactive compounds detection, constitute an integrated approach to maintain quality and safety of medicinal plants [1].

In this study, a DNA barcode molecular characterization was performed for the phylogenetic analysis of the *C. crassa* species from Ecuador and Peru using the ITS1 sequence, as well as their phytochemical analysis, which are essential aspects for the validation of the traditional use referred to for the plant.

## 2. Materials and Methods

### 2.1. Collection of Plant Material

Approval for collecting *C. crassa* samples was granted by the Ministry of the Environment of Ecuador with authorization code 022-2019-IC-FLO-DNB/MA.

Plants of *C. crassa* were collected in August 2018 from two different countries: Ecuador and Peru. Three plants were collected in La Libertad province, Santiago de Chuco Department Agasmarca, Peru (08°07′53″ S, 78°03′23″ W, 2900 m elevation) and two plants from the Yanachoca Reservation in North of Pichincha province, Ecuador (00°05′ S, 78°33′ W, 3700 m elevation). The collected plants were identified at the GUAY herbarium of Natural Science Faculty, Guayaquil University and deposited under the voucher specimen of 13.115 (*C. crassa* from Peru) and 13.116 (*C. crassa* from Ecuador). The GUAY herbarium performed the taxonomic analysis using morphological features of different tissues of the plants (accessed 13 December 2022: sweetgum.nybg.org/science/ih/herbarium-details/?irn=126275, www.gbif.org/es/grscicoll/institution/e1bd7b60-289b-471a-9b6a-bd9221cec189).

### 2.2. Phylogenetic Analysis

#### 2.2.1. DNA Extraction

Tissues collected from the two plants collected in Ecuador and the three plants collected in Peru. Tissue was ground using the MM400 equipment (Retsch, Hann, Germany) with liquid nitrogen and stored at −80 °C. Therefore, DNA extraction was performed on pulverized tissues from samples from two plants of *C. crassa* collected in Ecuador (codes CIBE-17 and CIBE-18: coordinates 00°05′ S, 78°33′ W) and three from Peru (codes CIBE-13, CIBE-14 and CIBE-15: coordinates 08°07′53″ S, 78°03′23″ W), using a modified CTAB protocol [13] by using an extraction buffer containing 2.8% CTAB, 1.3 M NaCl, 20 mM EDTA, 100 mM Tris–HCl pH 8, 1% PVP40000, 2% 2-mercaptoethanol. Approximately 100–150 mg of ground sample was used as starting material. The DNA was resuspended in 50 µL of TE buffer (10 mM Tris, 1 mM EDTA). The DNA was stored at −20 °C until it was used in the PCR reaction.

#### 2.2.2. PCR

PCR was performed using the primers 5aF (CCTTATCATTTAGAGGAAGGAG) and 4rev (TCCTCCGCTTATTGATATGC) according to Chen et al. [14]. The GoTaq^®^ 2× master mix (Promega, Madison, WI, USA) was used for PCR according to manufacturer’s instruction using 0.5 µM of each primer in a 30 µL PCR reaction volume. Each extracted DNA was amplified twice per sample (two technical replicates). PCR conditions were as followed: 95 °C for 3 min; 35 cycles of 95 °C for 30 s, 50 °C for 60 s, and 72 °C for 60 s; followed by a final extension of 72 °C for 5 min. Agarose gel electrophoresis was performed for amplicon detection by loading 5 µL of PCR reaction. The remaining 25 µL was sent for PCR purification and sequencing to Macrogen Inc. (Seoul, Republic of Korea).

#### 2.2.3. Bioinformatic Analysis

The two DNA sequences for each technical replicate were used to generate a consensus sequence for each biological replicate (two from Ecuador and three from Peru) after alignment using MUSCLE in MEGAX [15]. BLAST analysis was performed in the GenBank nr database [16] on 10 August 2021; therefore, the similarity search corresponded to accessions available in the database at the time of the analysis. Phylogenetic analysis was performed with different selected accessions based in the BLAST results. The different accessions from the GenBank and the sequences from the *C. crassa* from Ecuador and Peru were aligned in MEGAX using MUSCLE. The aligned sequences were trimmed at the ends to maintain the same range. The best model for phylogenetic analysis was used as recommended by MEGAX and the maximum likelihood method was performed by using a bootstrap test of 1000 replicates.

### 2.3. Phytochemical Analysis

#### 2.3.1. Preparation of Extracts

The whole plant was dried in an oven (AISET model VLD-6000, Yatai, Shanghai, China) with controlled temperature (40 °C ± 2 °C) for seven days, and following crushing was performed using a Pulvex mill to a particle size of 2 mm for processing and analysis. One hundred grams of each dry and ground plant previously wetted with 80% ethanol (Sigma-Aldrich, Darmstadt, Germany) was incubated for 15 min and transferred through a percolator. The extraction process was performed with 80% ethanolic solution, obtaining a fluid extract according to Miranda and Cuéllar [17]. The extracts were maintained at 10–15 °C for four days and then filtered.

#### 2.3.2. Gas Chromatography–Mass Spectrometry Analysis (GC-MS)

A portion of each hydroalcoholic extract was dried and mixed with N, O-Bis (trimethylsilyl) trifluoroacetamide (BSTFA), and heated in a water bath at 80 °C for 2 h to allow the silylation of metabolites [18]. Gas chromatography–mass spectrometry equipment (GC-MS) was performed using the equipment Agilent Technologies (7890A GC system and 5975C inert XL MSD with triple-axis detector). The capillary column DB-5MS (30 m × 0.25 mm) was used with phenyl dimethylpolysiloxane as stationary phase (0.25 µ film thickness) and helium gas (1.0 mL/min) as the carrier. One microliter of hydroalcoholic extracts was injected with the splitless mode at 250 °C. The temperature of the oven started at 70 °C for 2 min, then to 285 °C at 5 °C/min increment, and maintained at 285 °C for 2 min. The identification of compounds was performed by comparison of mass spectra based on Wiley 9th and NIST 2011 MS libraries. An electron ionization of 70 eV at 230 °C was performed in the ion source, and the data were collected with the full scan mode (40–600 amu) in the quadrupole mass analyzer.

## 3. Results

### 3.1. Phylogenetic Analysis

The ITS1 sequences from the different samples of *C. crassa* analyzed from Ecuador (GenBank accession numbers OM471920 and OM471919 for isolates CIBE-17 and CIBE-18, respectively) and from Peru (GenBank accession numbers OM471921, OM471922 and OM471923 for isolates CIBE-13, CIBE-14 and CIBE-15, respectively) showed low query coverage (between 24% and 43%, Table 1) for the blast similarity search results in the GenBank database to accessions corresponding to *Helosis* spp. The best model for phylogenetic analysis using the ITS1 sequences was Kimura 2-parameter with a discrete Gamma distribution, and with some sites been evolutionarily invariable (K2 + G + I). Phylogenetic analysis revealed that the ITS1 could be used to differentiate between *C. crassa* of Ecuador and Peru, showing two different clades between them with a bootstrap value of 100 (Figure 1). Furthermore, different clades were formed for other genera including *Helosis* spp. (family Balanophoraceae) and *Korthalsella* spp. (family Santalaceae), among others, indicating that the ITS1 sequence is useful to differentiate between family, genera, and species. Due to the low percentage identity in the blast analysis (83.97–92.22%, Table 1) with low query cover (24–42%, Table 1), analysis with closely related species could not be performed. On the other hand, analysis aimed to test if the ITS1 could group within each family clade the different genera, which was successful. As outgroup, different species belonging to different families and orders were used (Figure 1). At the family level, the phylogenetic tree showed the aggrupation of clades (Figure 1). Furthermore, the families compared in the phylogenetic tree belongs to different orders, while the family Balanophoraceae of the species studied (*C. crassa*) are clustered in a clade. All the genera of this family are parasitic plants, including *C. crassa*. The ITS1 sequence proved that at the different genera from each family tested were grouped in the phylogenetic tree.

### 3.2. Phytochemical Analysis

The chromatograms obtained by GC-MS system of the hydroalcoholic extracts from Ecuador and Peru are presented (Figure 2). The compounds identified are shown (Table 2). The compounds appear in order of elution of the chromatographic column. In general, differences of the relative abundances of the peaks among origins were observed.

## 4. Discussion

The use of DNA barcodes is an important tool as a complement analysis for the identification of medicinal plants. Together with phytochemical analysis including chromatographic techniques such as GC-MS, DNA barcoding could be used for proper species identification and quality control of medicinal products. Most plants may have considerable differences in morphology even if they belong to same genus or species; for instance, at morphological level, the *C. crassa* from Ecuador showed a more pronounced contour of the epidermal cells of the powdered drug when compared to the *C. crassa* from Peru [9]. Therefore, DNA analysis could be useful as a complement analysis for species identification and even discriminate between genuine and adulterated products [20]. DNA barcodes are defined as the short segment of the gene sequence that evolves radically to differentiate species [21,22]. DNA barcoding uses short regions of DNA to distinguish between species and could be used for the authentication of medicinal plants and their products [12,23,24]. Most medicinal plants contain chloroplasts; and therefore, some DNA sequences from the plastid genome are recommended as universal DNA barcodes for phylogenetic analysis and species identification in plants [25]. For instance, the *rbcL* and *matK* is recommended as a 2-*locus* barcode for species identification [25]. However, parasitic plants may have reconfigured the plastome or even lost photosystem and energy production genes due to relaxation pressure as they rely on plant host for crucial function [26,27,28,29]. In this case, DNA sequences from mitochondrial or nuclear genome could be used for DNA barcoding. Therefore, for parasitic plants, other barcodes than from chloroplast origin could be used for successful PCR amplification and/or sequence analysis. The ITS sequence has been used in different organisms for phylogenetic analysis and species identification, including fungus [30] and plants [31]. The ITS sequence is the fastest evolving region of rRNA in many organisms [32]. Furthermore, in medicinal plants, the ITS sequences (ITS1 and/or ITS2) showed a better resolution for species identification or differentiation than other barcodes, like the *rbcL* or *matK* [33,34,35,36,37]. Blast results showed no significant similarities (at least 80% of query cover with >95% of the sequence similarity). This result is due to the fact that no ITS sequences for *C. crassa* are available in the GenBank database at the time of the analysis. Phylogenetic analysis of the ITS1 sequences revealed that this barcode could be used to differentiate *C. crassa* from Ecuador and Peru. This result is in accordance with other studies indicating that the ITS sequence could be used for species identification or differentiation in medicinal plants [33,34,35,36,37]. The DNA barcode analysis using the ITS1 sequences has not been reported before to genotype the species of the genus *Corynaea* from different origins, which constitutes a novel aspect in the identification of the plant.

The phytochemical analysis by GC-MS of the plants collected in Ecuador and Peru revealed the presence of 16 compounds for each hydroalcoholic extract. The fundamental difference was in the relative abundance of the similar compounds identified, and others that prevailed in one extract and not in the other. The majority compound in both fractions was 2,4-dimethylbenzaldehyde, although in the Peruvian sample the percentage was higher than in the Ecuadorian. The 2,4-dimethylbenzaldehyde is an aromatic compound present in the essential oil of several medicinal plants such as *Cassia siamea* and *Cassia occidentalis* (Fabaceae), *Cnestis ferruginea* Vahl ex DC (Connaraceae) [38] (and *Albizia adiantifolia* (Fabaceae) [39]. Furthermore, it is used in perfume, soap, food and drink production, and as solvent for resins and oils [40]. This compound has not been previously identified in the genus *Corynaea* or the family Balanophoraceae. Other major compounds but with differences in relative abundances in both extracts were stearic acid, phosphoric acid and hexadecanoic acid. The compounds that made the difference were eugenol, 1,4-lactone arabinonic acid, and dimethoxyrabelomycin, which were identified in the Ecuadorian sample and not in the Peruvian one, and azelaic acid, protocatechuic acid, and 9-octadecenamide which were detected in the Peruvian sample but not in the Ecuadorian one. Furthermore, eugenol, 1,4-lactone arabinonic acid, dimethoxyrabelomycin, and azelaic acid are reported for the first time for the *C. crassa* species and genus. Previous phytochemical studies performed by López et al. [8] to a fraction of ethyl acetate obtained from the drug, also revealed differences. In *C. crassa* from Ecuador, the compounds safrole and squalene were identified as major components, while the hexadecanoic and octadecanoic acids for the Peruvian species. The differences found in the species from different origins may be due to the geographical ecological conditions of the place where they grow and their condition as a parasitic plant [41]. The latter affects the appearance of compounds that are not common for the genus, while the quantitative differences may be dependent on extrinsic factors. The habitat could be a factor affecting the amount and accumulation of secondary metabolites. The location where a plant grows could affect the production process of effective substances due to changes in temperature, humidity, characteristics and elevation of the site and soil, and soil pH, among other factors [42,43].

## 5. Conclusions

Genetic and phytochemical analysis were performed in *C. crassa* from Ecuador and Peru. Differences were detected in Ecuadorian and Peruvian samples of *C. crassa* for the DNA barcoding using the ITS1 sequence and for the phytochemical analysis using GC-MS. Therefore, genetic and phytochemical analysis could be used in *C. crassa* to differentiate from different origins and could be used in quality control in medicinal products that use *C. crassa*.

## Figures and Tables

**Figure 1 genes-14-00088-f001:**
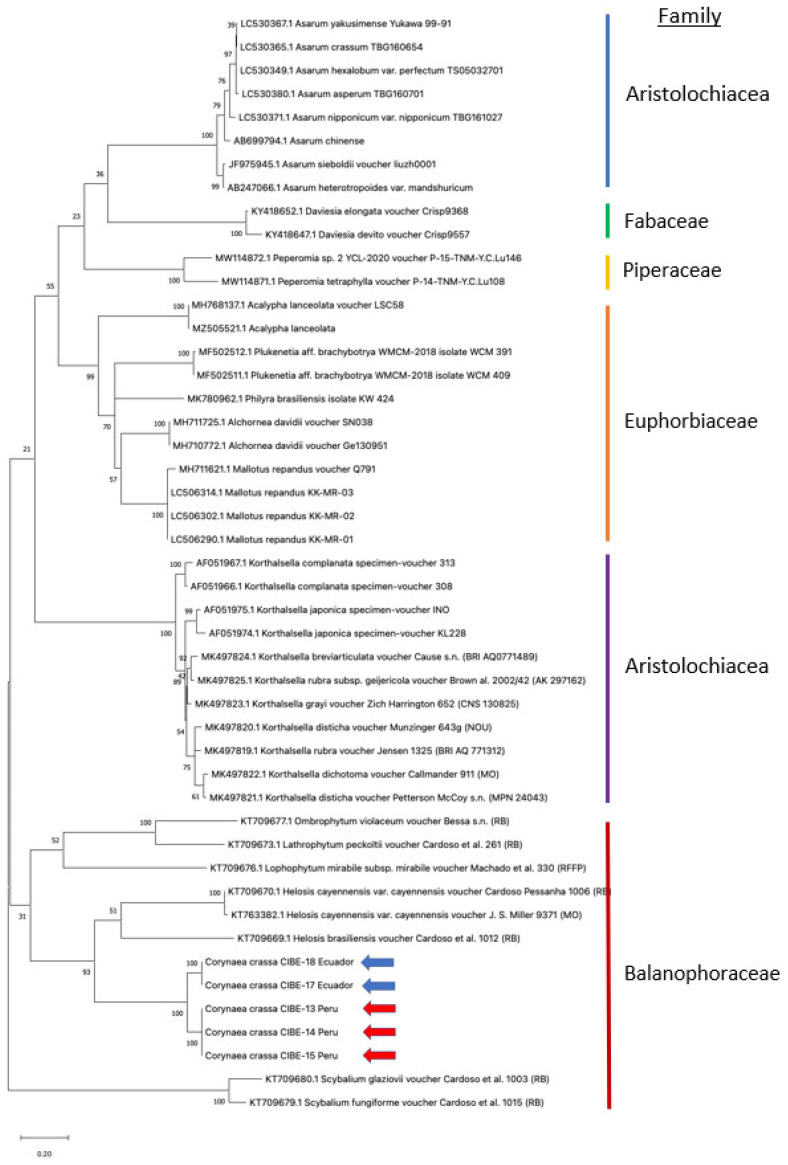
Phylogenetic analysis was performed by using the Maximum Likelihood method and Kimura 2-parameter model of the ITS1 sequences from the *C. crassa* from Ecuador and Peru, and from accessions queried in the GenBank database. The percentage of trees in which the associated taxa clustered together is shown next to the branches. A discrete Gamma distribution was used to model evolutionary rate differences among sites (5 categories (+*G*, parameter = 1.7045)). The rate variation model allowed for some sites to be evolutionarily invariable ((+*I*), 19.64% sites). The tree is drawn to scale, with branch lengths measured in the number of substitutions per site. This analysis involved 47 nucleotide sequences. There was a total of 673 positions in the final dataset. Evolutionary analyses were conducted in MEGA X [15,19]. Blue and red arrows indicate *C. crassa* from Ecuador and from Peru, respectively.

**Figure 2 genes-14-00088-f002:**
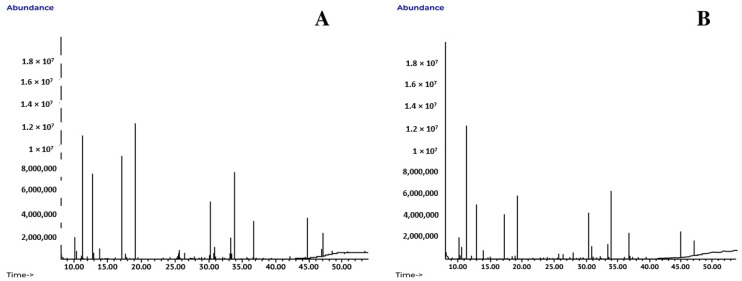
Gas chromatograms of the hydroalcoholic extracts of *C. crassa* collected in Ecuador (**A**) and Peru (**B**).

**Table 1 genes-14-00088-t001:** Blast analysis of the ITS1 sequences from the *C. crassa* of Ecuador and Peru.

		BLAST				
*C. crassa* Source	Code	Organism	Accession	Query Cover (Query Length)	E-Value	Identity
Ecuador	CIBE-18	*Helosis cayennensis* (Sw.) Spreng	**KT709670**	24% (676 bp)	2 × 10^−56^	92.22%
Ecuador	CIBE-17	*H. cayennensis* (Sw.) Spreng	**KT709670**	24% (677 bp)	2 × 10^−56^	92.22%
Peru	CIBE-13	*Helosis brasiliensis*	**KT709669**	39% (715 bp)	6 × 10^−66^	83.97%
Peru	CIBE-14	*H. brasiliensis*	**KT709669**	43% (651 bp)	6 × 10^−66^	83.97%
Peru	CIBE-15	*H. brasiliensis*	**KT709669**	42% (664 bp)	6 × 10^−66^	83.97%

**Table 2 genes-14-00088-t002:** Compounds identified in the hydroalcoholic extract from *C. crassa* collected in Ecuador (GenBank accession number OM471920) and Peru (GenBank accession number OM471921).

	RT (min)	Compounds	RA (%) *	
			Ecuador	Peru
1	11.34	2,4-Dimethylbenzaldehyde	11.98/0.50	18.07/0.88
2	12.06	Benzoic acid	0.39/0.05	0.45/0.06
3	12.74	Phosphoric acid	6.21/0.06	5.86/0.80
4	13.91	Butanedioic acid	0.99/0.06	0.96/0.13
5	17.83	Eugenol	0.53/0.03	-
6	21.73	Arabinonic acid, 1,4-lactone	0.17/0.04	-
7	25.34	Azelaic acid	-	0.24/0.03
8	25.69	D-Psicofuranose	1.03/0.07	0.66/0.04
9	25.79	Protocatechuic acid	-	0.31/0.02
10	26.43	Tetradecanoic acid	0.79/0.04	0.74/0.06
11	28.57	Gallic acid	0.54/0.12	0.40/0.14
12	30.30	Hexadecanoic acid	5.21/0.14	5.95/0.43
13	30.88	Oleanitrile	1.18/0.09	1.76/0.12
14	33.24	9,12-octadecadienoic acid	0.99/0.17	0.70/0.12
15	33.36	Oleic acid	2.29/0.25	2.19/0.27
16	33.49	11-cis-octadecenoic acid	0.53/0.03	0.60/0.05
17	33.85	Octadecanoic acid	7.68/0.24	8.34/0.64
18	36.61	9-octadecenamide	-	3.70/0.33

* The results are expressed as medium/standard deviation (*n* = 3). RT: Retention time expressed in minutes; RA: Relative abundance expressed as peak area percent.

## Data Availability

DNA sequences are available at the GenBank (www.ncbi.nlm.nih.gov/genbank) under accessions numbers: OM471920, OM471919, OM471921, OM471922 and OM471923.

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
