# Peer review of "ITS1 Barcode and Phytochemical Analysis by Gas Chromatography–Mass Spectrometry of Corynaea crassa Hook. f (Balanophoraceae) from Ecuador and Peru"

_genes, 2022, doi:10.3390/genes14010088_

Round 1

Reviewer 1 Report

Dear Editor in Chief;

Journal of Genes:

Prof. Dr. Mia Gao

The manuscript entitled ITS1 barcode and phytochemical analysis by Gas Chromatography – Mass spectrometry of Corynaea crassa Hook. f (Balanophoraceae) from Ecuador and
Peru has been reviewed.  Accordingly, this manuscript needs to be revised (Major Revision).

Sincerely Yours

Author Response

13th December 2022

Dear Editor and reviewers of MDPI Gene

I would like to thanks the valuable comments of the editor and reviewers that enhance the quality of the manuscript. Please find the responses for every comments of the reviewers

REVIEWER 1

Comments:

Line 30: The accessions numbers were added in the abstract “Blast analysis was performed in the GenBank database with the ITS1 sequences obtained from two accessions of C. crassa from Ecuador (GenBank accession numbers OM471920 and OM471919 for isolates CIBE-17 and CIBE-18, respectively) and three from Peru (GenBank accession numbers OM471921, OM471922 and OM471923 for isolates CIBE-13, CIBE-14 and CIBE-15, respectively).”

Line 35: The words “different locations” were added.

Line 58: The word “flowering” was added instead of “inflorescence”.

Line 67: The sentence was maintained as it refers to another article López et al. 2021) which published hydroalcoholic extracts: The antioxidant activity was determined with hydroalcoholic and water extracts. On the other hand, in López et al. (2020) fractionation was performed employing different solvents with increasing polarity: hexane, ethyl acetate, ethanol for the phytochemical study.

Lines 97-99: The sentence was modified:

“Plants of C. crassa were collected in August 2018, from two different countries: Ecuador and Peru. Three plants were collected in La Libertad province, Santiago de Chuco Department Agasmarca, Peru (08°07′53″S, 78°03′23″W, 2900 m elevation) and two plants from the Yanachoca Reservation in North of Pichincha province, Ecuador (00°05′S, 78°33′W, 3700 m elevation). The collected plants were identified at the GUAY herbarium of Natural Science Faculty, Guayaquil University and deposited under the voucher specimen of 13.115 (C. crassa from Peru) and 13.116 (C. crassa from Ecuador). The GUAY herbarium performed the taxonomic analysis using morphological features of different tissues of the plants[1], [2].”

The use of separated accessions was to discover if the ITS1 sequence showed similarity or differences between two different locations (Ecuador and Peru) of the same specie; and how it was located in phylogenetic tree analysis when compared to other published Genbank accessions.

Line 102: Paragraph was modified: Tissues collected from the two plants collected in Ecuador and the three plants collected in Peru. Tissue was ground using the MM400 equipment (Retsch, Hann, Germany) with liquid nitrogen and stored at -80°C. Therefore, DNA extraction was performed on pulverized tissues from samples from two plants of C. crassa collected in Ecuador (codes CIBE-17 and CIBE-18: coordinates 00°05′S, 78°33′W) and three from Peru (codes CIBE-13, CIBE-14 and CIBE-15: coordinates 08°07′53″S, 78°03′23″W), using a modified CTAB protocol (Pacheco et al., 2017).

Line 135: The whole plant was used: “The whole plant was dried in an AISET model VLD-6000 (China) oven…”

Line 164: When performing blast analysis, the software looks for DNA sequence similarity on the GenBank nucleotide database. The accessions with higher similarity were from Helosis spp. Therefore, this genus was used for phylogenetic analysis.

Line 170: This genus was used because of the similarity rate using blast (see above), the family names was indicated for this specie as well for Helosis

Table 1: The author for the taxon is (Sw) Spreng. The name was added to the table

Figure 1, line 173: according to Kuma Ojha et al. 2022[3]: “Bootstrap values in a phylogenetic tree indicate that out of 100, how many times the same branch is observed when repeating the generation of a phylogenetic tree on a resampled set of data. If we get this observation 100 times out of 100, then this supports our result.”

Line 173:

Why did the authors analyze other genera such as Alchornea, Mallotus, Acalypha, Peperomia and Asarum?

The phylogenetic tree should show the aggrupation of clades according to Families. That’s why different genera were queried for ITS sequence that belong to different families. The Figure was modified showing the families.

What are the taxonomic relationships or values between these taxa and the species studied in this research?

The families compared in the phylogenetic tree belongs to different orders; while the Family (Balanophoraceae) of the species studied (Corynaea crassa) are clustered in a clade. All the genera of this Family are parasitic plants, including C. crassa.

What is the validity of this method regarding the phylogenetic approach? 

The ITS1 sequence prove that at the different genera from each Family tested were grouped.

What are the criteria to select the out groups? It needs to be revised and explained.

Due to the low percentage identity in the blast analysis (83.97%-92.22%) with low query cover (24%-42%) analysis with closely related species could not be performed. On the other hand, analysis was aimed to test if the ITS1 could group within each Family clade, the different genera; which was successful. As outgroup, different species belonging to different Families and Orders were used.

Therefore, after line 171 the following paragraph was added:

Due to the low percentage identity in the blast analysis (83.97%-92.22%, Table 1) with low query cover (24%-42%, Table 1) analysis with closely related species could not be performed. On the other hand, analysis was aimed to test if the ITS1 could group within each Family clade the different genera, which was successful. As outgroup, different species belonging to different Families and Orders were used (Fig. 1). At the Family level, the phylogenetic tree showed the aggrupation of clades (Fig. 1). Furthermore, the Families compared in the phylogenetic tree belongs to different orders; while the Family (Balanophoraceae) of the species studied (Corynaea crassa) are clustered in a clade. All the genera of this Family are parasitic plants, including C. crassa. The ITS1 sequence prove that at the different genera from each Family tested were grouped.

Table 2: GenBank accessions numbers were added in the text.

Table 2: In this study we analyzed independently the plants of C. crassa from Ecuador and Peru, at the genetic (ITS1 sequence) and phytochemical analysis (from ethanol extracts followed by GC-MS analysis). Both results indicate difference in both plants, which belongs to the same species but from different locations; therefore, differences we encountered in both methodologies. Therefore, a genotypic (ITS1) and phenotypic (GC-MS) differences were found.

Line 195: The paragraph was modified: The chromatograms obtained by GC-MS system of the hydroalcoholic extracts from Ecuador and Peru are presented in Figure 2. The compounds identified are shown in Table 2 and they appear in order of elution of chromatographic column. In general, differences of the relative abundances of the peaks among origins were observed.

The Table 2 was also modified for better comparisons between C. crassa from Ecuador and Peru

Line 201: The morphological analysis of C. crassa from Ecuador and Peru were described by López et al. 2021. At the microscopic level of the powdered drug, differences were encountered in the contour of the epidermal cells in which the C. crassa from Ecuador was more pronounced than the C. crassa from Peru. A sentence was added: At morphological level, the C. crassa from Ecuador showed a more pronounced contour of the epidermal cells of the powdered drug when compared to the C. crassa from Peru (López et al. 2021).

Other modifications:

Affiliation: line15, a misspelled was corrected: Biochemistry.

Changes in the text was performed according to the similarity report.

All changes could be tracked

Kind regards

Efrén Santos-Ordóñez

[1] sweetgum.nybg.org/science/ih/herbarium-details/?irn=126275

[2] www.gbif.org/es/grscicoll/institution/e1bd7b60-289b-471a-9b6a-bd9221cec189

[3]www.sciencedirect.com/topics/medicine-and-dentistry/bootstrapping#:~:text=Bootstrap%20values%20in%20a%20phylogenetic,then%20this%20supports%20our%20result.

Reviewer 2 Report

Dear Authors,

I was asked by Editors to review your manuscript entitled  "ITS1 barcode and phytochemical analysis by Gas Chromatography-Mass spectrometry of Corynaea crassa Hook. f (Balanophoraceae) from Ecuador and Peru". After reviewing your manuscript, my recommendation minor revision.

Line 17 - is this the correct formatting for an abstract, accepted by the journal? Please, check the formatting of other subsections as well.

Line 76 - I would suggest defining DNA barcoding at this point in the text, and not just in the Discussion. Just a succinct one-sentence explanation would suffice.

Line 174 - in Figure 1, the arrows supposedly indicate the closest taxa. I suggest that this should be explained in the caption following Fig. 1. 

Line 185 - Please, explain in more detail the differences in abundance. Which differences were observed? Etc.

Line 220 - why are ITS sequences better for DNA barcoding of medicinal plants compared to other genes? Please, explain in one or two sentences.

Author Response

13th December 2022

Dear Editor and reviewers of MDPI Gene

I would like to thanks the valuable comments of the editor and reviewers that enhance the quality of the manuscript. Please find the responses for every comments of the reviewers

Reviewer 2

Comments:

Line 17: Abstract was modified according to instructions of the journal.

Line 76: DNA barcode definition was added: DNA barcode is a short DNA sequence which is used for identification of species (Hebert et al. 2003) that could be used for rapid, accurate tool as a complement for taxonomic identification.

Line 174: The arrows in the figure are explained in the caption of the Figure.

Line 185: Paragraph was modified.

Line 220: The use of ITS is explained: The ITS sequence is the fastest evolving region of rRNA in many organism (White 1990).

Other modifications:

Affiliation: line15, a misspelled was corrected: Biochemistry.

Changes in the text was performed according to the similarity report.

All changes could be tracked

Kind regards

Efrén Santos-Ordóñez
